# Assessment of facility-based tuberculosis data quality in an integrated HIV/TB database in three South African districts

Joshua P. Murphy[1]*, Sharon Kgowedi[1], Lezanie Coetzee[1], Vongani Maluleke[1], Daniel Letswalo[1], Constance Mongwenyana[1], Pria Subrayen[2], Salome Charalambous[2,3], Lindiwe Mvusi[4], Sicelo Dlamini[4], Neil Martinson[5], Aneesa Moolla[1], Jacqui Miot[1], Denise Evans[1]

1 Health Economics and Epidemiology Research Office (HE²RO), University of the Witwatersrand, Johannesburg, South Africa, 2 The Aurum Institute, Parktown, Johannesburg, South Africa, 3 School of Public Health, University of the Witwatersrand, Johannesburg, South Africa, 4 South African National Department of Health, Pretoria, South Africa, 5 Perinatal HIV Research Unit (PHRU), Soweto, South Africa

* joshpmurphy@gmail.com

**Data Availability Statement:** The datasets generated and/or analyzed during the current study is owned by the South African Government and use

## Abstract

### Background

Assessment of data quality is essential to successful monitoring & evaluation of tuberculosis (TB) services. South Africa uses the Three Interlinked Electronic Register (TIER.Net) to monitor TB diagnoses and treatment outcomes. We assessed the quality of routine programmatic data as captured in TIER.Net.

### Methods

We reviewed 277 records from routine data collected for adults who had started TB treatment for drug-sensitive (DS-) TB between 10/2018-12/2019 from 15 facilities across three South African districts using three sources and three approaches to link these (i.e., two approaches compared TIER.NET with the TB Treatment Record while the third approach compared all three sources of TB data: the TB treatment record or patient medical file; the TB Identification Register; and the TB module in TIER.Net). We report agreement and completeness of demographic information and key TB-related variables across all three data sources.

### Results

In our first approach we selected 150 patient records from TIER.Net and found all but one corresponding TB Treatment Record (99%). In our second approach we were also able to find a corresponding TIER.Net record from a starting point of the paper-based, TB Treatment Record for 73/75 (97%) records. We found fewer records 55/75 (73%) in TIER.Net when we used as a starting point records from the TB Identification Register. Demographic information (name, surname, date of birth, and gender) was accurately reported across all three data sources (matching 90% or more). The reporting of key TB-related variables was

is governed by the University of the Witwatersrand Human Research Ethics Committee (HREC – Medical). Data cannot be shared publicly because the local ethics committee restricts access. Access to these data is subject to restrictions owing to privacy and ethics policies set by the South African Government, so they are not publicly available. Requests can be directed to: Health Economics and Epidemiology Research Office (HE2RO) Ross Greener rgreener@heroza.org HE2RO's Head of Data or information@heroza.org. Human Research Ethics Committee (HREC) at the University of Witwatersrand Professor Clement Penny, +27 11 717 2301, or by e-mail on Clement.Penny@wits.ac.za. The telephone numbers for the Committee secretariat are +27 11 717 2700/1234 and the e-mail addresses are Zanele.Ndlovu@wits.ac.za and Rhulani.Mukansi@wits.ac.za.

**Funding:** This study was funded by the Bill and Melinda Gates Foundation (OPP 1173131) under through the South African National TB Think Tank and Aurum Institute. https://www.gatesfoundation.org/ The contents of this publication are solely the responsibility of the authors and do not necessarily represent the official views of the National TB Think Tank or its funders. This work has also been made possible by the generous support of the American People and the President's Emergency Plan for AIDS Relief (PEPFAR) through USAID under the terms of Cooperative Agreements 72067419CA00004 to HE2RO. The contents are the responsibility of the authors and do not necessarily reflect the views of PEPFAR, USAID or the United States Government. https://www.state.gov/pepfar/ Research reported in this publication was also supported by the Fogarty International Center and National Institute of Mental Health, of the National Institutes of Health under Award Number D43 TW010543. The content is solely the responsibility of the authors and does not necessarily represent the official views of the National Institutes of Health. https://sites.sph.harvard.edu/global-health-research-partnership/ The funders had no role in the study design, collection, analysis and interpretation of the data, in manuscript preparation or the decision to publish.

**Competing interests:** The authors declare no competing interests.

similar across both the TB Treatment Record and the TB module in TIER.Net (p>0.05). We observed differences in completeness and moderate agreement (Kappa 0.41–0.60) for site of disease, TB treatment outcome and smear microscopy or X-ray as a diagnostic test (p<0.05). We observed more missing items for the TB Treatment record compared to TIER.Net; TB treatment outcome date and site of disease specifically. In comparison, TB treatment start dates as well as HIV-status recording had higher concordance. HIV status and lab results appeared to be more complete in the TB module in TIER.Net than in the TB Treatment Records, and there was "good/substantial" agreement (Kappa 0.61–0.80) for HIV status.

## Discussion and conclusion

Our key finding was that the TB Module in TIER.Net was more complete in some key variables including TB treatment outcome. Most TB patient records we reviewed were found on TIER.Net but there was a noticeable gap of TB Identification patient records from the paper register as compared to TIER.Net, including those who tested TB-negative or HIV-negative. There is evidence of complete and "good/substantial" data quality for key TB-related variables, such as "First GeneXpert test result" and "HIV status." Improvements in data completeness of TIER.Net compared to the TB Treatment Record are the most urgent area for improvement, especially recording of TB treatment outcomes.

## Introduction

The World Health Organization's (WHO) "Joint Review of HIV, TB and PMTCT Programs in South Africa" (2014), as well as more recent reports from UNAIDS (2018), emphasize that integration of HIV and tuberculosis (TB) services are necessary for adequate service delivery at primary health care facilities in South Africa [1–3]. Quality TB data is also critical for reporting [1–5], a necessary condition to conduct monitoring and evaluation (M&E) of programs and to inform decisions on policy and allocation of scarce resources [4, 6]. Inaccurate estimates at the facility level mean inaccurate estimates of the burden of TB disease at the national level. The quality of these "on the ground" data need to be understood and improved in order to ensure the quality of data reported for WHO's TB country profile and Global Tuberculosis Report are of adequate quality [7].

### TIER.Net and ETR.Net

The Three Interlinked Electronic Register (TIER.Net) was introduced in South Africa in December 2010 and has been rolled out to the majority of public-sector ART service points [8]. TIER.Net is an offline (networked sometimes within a single facility) electronic program designed to maintain TB/HIV patient information [8]. It is described as a patient management database and a tool to support routine collection and reporting of programmatic data into the webDHIS which is a flexible, web-based open-source health information system [9]. The University of Cape Town's (UCT) Centre for Infectious Diseases, Epidemiology and Research (CIDER) and other partners developed and supported the roll out of this electronic medical record software [10].

TIER.Net provides reports of patients being treated for HIV on a monthly and quarterly basis (8). The TB module in TIER.Net was designed to capture patient level TB data

electronically so that integrated TB and HIV information could be immediately accessible and used for improving client care at facility level. The electronic TB register (ETR.Net) preceded the TB module in TIER.Net and was warehouse of TB data since 2004; recognized as a valuable tool to guide program management of TB services [11]. ETR.Net was decommissioned around March 2019, in support of an integrated TB and HIV information system. The TB module of TIER.Net was rolled out in 2016 but, to date, this process has only been evaluated once with an emphasis on implementation and fears around quality during the transition from ETR.Net to TIER.Net [12].

## Quality of TB data

What we know about TB data quality in South Africa is limited to ETR.Net and the Electronic Drug Resistant software (EDRWeb), the latter of which is used for drug-resistant TB and beyond our focus [11, 13, 14]. In short, ETR.Net was described as user-friendly, secure and flexible; however it was not based at the facility-level and relied on a combination of paper-based registers [15]. ETR.Net was able to achieve its objective of informing the drug-sensitive (DS) TB programme, but it was not without concerns of completeness and error [11, 13]. Quality TB data are a priority of the South Africa National Department of Health (NDoH); South Africa's National Strategic (NSP) for HIV, TB and STIs 2017–2022 (NSP 2017–2022) [16] is aligned to international targets from the WHO including the Global End TB Strategy by 2030 [5]. The NSP calls specifically for 90-90-90 targets for TB that: 90% of all people who need TB treatment are diagnosed and receive appropriate therapy as required; 90% of people in key and vulnerable populations are diagnosed and receive appropriate therapy; and treatment success is achieved for least 90% of all people [16]. For these targets to be measured and for the TB-care cascade to be effectively monitored, successful integration of the TB module within TIER.Net and achieving a minimum level of data quality is of paramount importance. We therefore assessed the quality of TB and TB-related data captured in TIER.Net.

## Methods

We conducted this data quality assessment of the TB module in TIER.Net to understand the strengths and weaknesses of the TB module in three districts of South Africa: Johannesburg Metro in Gauteng Province (GP), Mopani in Limpopo province (LP) and Bojanala Platinum in the North West (NW) province of South Africa. Districts were selected based on consultation with the South African NDoH Drug-sensitive (DS)-TB Directorate and National TB-Think Tank as well as accessibility and presence of established data collection teams operating in Johannesburg (within GP and near NW) and Tzaneen (within LP). Districts and facilities were selected based on the following criteria: (1) TB cases diagnosed and treatment start rate, identified from the routine District Health Information System (DHIS) and District Health Barometer 2018/2019 [17]. We selected one high volume province (Gauteng) and two low volume provinces (North West and Limpopo). Within these, we selected one district above the treatment start rate (Bojanala Platinum 108.0 versus SA average 97.8) and one below the treatment start rate (Mopani 95.2 versus SA average 97.8). Johannesburg was below the average at 89.2., (2) the environment, including urban (Johannesburg), per-urban (Bojanala Platinum) or rural (Mopani), (3) whether we had existing relationships with facilities or districts, and last (4) whether we had staff that was familiar with these areas and could move through these areas easily. Data collectors extracted data from 15 facilities between 15 October and 15 December 2020.

## Study population

We reviewed routine data collected from adult patients that started TB treatment for DS-TB between October 2018 and December 2019. We allowed a minimum of 6 months of follow-up time to ensure that the full record of TB treatment with outcomes was available to review. Records were collected from three sources (Table 1) and were systematically selected using three sources (Fig 1). Eligibility criteria included: aged 18 years or older, DS-TB only and (not extensively drug-resistant TB nor multidrug-resistant tuberculosis), and a TB treatment start date in the timeframe above. Records were collected consecutively, according to the abovementioned criteria and the procedures outlined under "Data Sources", and working backward until the required number of records had been collected.

## Data sources

We used three data sources for this data quality assessment of the TB module in TIER.Net; two paper (i) The TB Identification Register, (ii) TB Treatment Record or patient medical file, and one electronic (iii) TB module in TIER.Net. Three different approaches were used to link the different data sources. The first two compared TIER.Net with the TB Treatment Record while the third approach compared all three sources of TB data: The TB Identification Register, TIER.Net and the TB Treatment Record. The three approaches followed these sampling steps (also described in Fig 1): 1) at each facility (n = 15), ten patients were selected from the TB module in TIER.Net by filtering the electronic records in the database by last "TB Start Date", from the first record closest to a 30 September 2019 TB start date and then working back in

**Table 1. Description of data sources.**

| Source | Description |
|---|---|
| (i) The TB Identification Register (a paper-based register maintained by each TB treatment facility) | The entry from routine screening into TB diagnostic testing. The purpose of the Register is to collect necessary information on people with TB symptoms, and assist with the following: |
| | • follow-up of patients with positive results who do not come back for their results; |
| | • monitoring whether all the results of specimens sent to the laboratory are returned to the facility; |
| | • monitoring of the Turn-Around-Time (TAT) for results; |
| | • estimating the laboratory supplies needed by the facility; |
| | • follow-up of symptomatic patients referred to hospital for further investigations and final diagnosis. |
| | This replaced the TB Suspect Register and the latest version is from 2020, Form code: GW20/13 (S1 File). Also note the paper TB Treatment Register is no longer used. |
| (ii) TB Treatment Record (paper-based/patient medical file) | This is the blue paper record that is either kept separately in a box or filing cabinet with other TB records or placed inside the brown folder, the patient medical record. The purpose of the TB Treatment Record is to record patients' TB clinical treatment history including testing, treatment and outcomes. The latest version is from 2018, Form code: GW 20/12 (S2 File). |
| (iii) TB module in TIER.Net (electronic) | This is the electronic TB record that includes information from source (i) and (ii) as well as relevant HIV information and basic demographics (such as names, gender, and date of birth) |
| | During our data collection facilities transitioned from version 1.13.2 to 1.13.3 around Oct/Nov 2021. (S1 Table) |

| | 1) TIER.Net to patient medical file | 2) Patient medical file to TIER.Net | 3) TB Identification Register to TIER.Net |
|---|---|---|---|
| Source |  |  |  |
| Records per facility | 10 | 5 | 5 |
| Justification | This was the most convenient approach to find records in a systematic and reliable way. | The aim was to confirm that archived records were consistently recorded in TIER.Net. Here we also had an opportunity to check patient records that were more likely TB only as opposed to HIV/TB patients. | This route was important for us to understand the entire data pathway from testing/registration to treatment. We planned to collect 2 records with TB treatment start dates and 3 who tested TB negative, i.e. not TB treatment start dates. |
| Sampling frame | Started with the last TB treatment start date in September 2019 and worked backwards in time screening every fifth record. If that record was not eligible we would take the next record and continue the process of screening every fifth | Aimed to find a record from each calendar quarter (Oct-Dec 2018, Jan-Mar 2019, Apr-Jun 2019, Jul-Sep 2019 Oct-Dec 2019) selecting the first record starting from the earliest date of each quarter. | |
| Total number of records from 15 facilities | 150 | 75 | 75 |

**Fig 1. Record targets and route of identification.**

time selecting every 5th record then finding the corresponding TB Treatment Record in the facility's filing room; 2) five TB Treatment Records which met the eligibility criteria were identified from available clinical records and then paired with the electronic record in the TB module in TIER.Net; 3) lastly five records identified from the TB Identification Register were then paired with the TB Treatment Record as well as the electronic record in the TB module in TIER.Net.

## Data collection and management

We piloted data collection tools at a single facility not included in the 15 selected study sites. Study tools were revised and finalised before data collection started. Data collectors were trained in Good Clinical Practice, research ethics, and study procedures prior to commencing data collection. Study staff with experience in reviewing HIV/TB clinical records were trained to extract key data from the data sources and enter this into REDCap [18]. We conducted ad-hoc quality checks to reduce missing data and errors in data entries. Data were entered directly

into REDCap using laptops on site. Department of Health (DoH) facility staff assisted the study team with locating TB Treatment Records and the TB Identification Register and accessing the TB module in TIER.Net on facility computers. National Health Laboratory System (NHLS) laboratory results were considered a part of the TB Treatment Record but only if printed and included in patients' medical records.

## Data analysis

Stata version 15 was used for analysis. Data were summarized using standard descriptive statistics [19]. We created two datasets; the first included data from the TB module in TIER.Net and the TB Treatment Record, and the second focused on the TB Identification Register. For the second, we could only assess completeness and agreement for a small subset of indicators in the TB Identification Register. This limited the inclusion of this source in some of the comparisons outlined below.

First, to describe the sample, we used proportions for the characteristics of the sample (e.g., records found, gender and route of identification), stratified by district. Second, to describe the accuracy of demographic information, we present the proportion of records that matched (agreed) for each variable, stratified by district, for the different identification routes. A match was identified if both sources of information reported identical information. For the purpose of this analysis, we excluded records if the corresponding record could not be located (i.e., TIER.Net to TB Treatment Record; TB Treatment Record to TIER.Net, or TB Identification Register to either TIER.Net or TB Treatment Record). We also present data where the variable of interest (e.g., name, surname, date of birth, gender, folder number) was present in both sources and could be compared. Minor spelling differences were considered matches. Proportions were compared using the chi-square test.

Third, to describe the completeness of data, we compared the reporting of key TB-related variables across TIER.Net and the TB Treatment Record. TB treatment outcomes were presented according to WHO's standard definitions and reporting framework for TB as cured, completed treatment (treatment success (the sum of cured and treatment completed), died, lost to follow up, treatment failure and not evaluated) [20]. Again, proportions were compared using the chi-square test.

Fourth, to compare the accuracy of key TB-related variables and outcomes, we used agreement between measurements. We used the weighted Kappa statistic for ordinal data, where there are more than two categories assessed by two techniques. We present the observed agreement (by calculating the frequency with which the two measurements agreed), the expected agreement (first calculating the expected values of the cells in the 2×2 table using the marginal frequencies, then using those cell numbers to calculate the frequency with which the two measurements are expected to agree) and the Kappa (if the two measurements agree) [21]. The formula for Kappa is (observer agreement—expected agreement)/1- expected agreement. The value of Kappa was used to determine the strength of agreement; <0.20 poor, 0.21–0.40 fair, 0.41–0.60 moderate, 0.61–0.80 good/substantial and 0.81–1.00 very good/almost perfect [21, 22].

Last, to demonstrate if laboratory results agreed, we compared 2x2 tables with the proportion of patients with a positive or negative first GeneXpert or first smear result, obtained from the clinic record or TIER.Net. This analysis was restricted to patients who had a first GeneXpert or first smear result recorded in the clinic record and in TIER.Net. Then, we repeated this for the proportion of patients that had a successful or unsuccessful TB treatment outcome. Treatment outcomes were re-classified into successful (sum of cured and treatment completed) and unsuccessful TB treatment outcome (composite outcome including loss to follow-

up, died, and treatment failure). Missing outcomes (no treatment outcome was assigned) or where the outcome was "moved/transferred out to another treatment unit" were also included as unsuccessful outcomes. Time to TB treatment outcome was calculated from the start of TB treatment until the date the outcome was assigned, where the TB treatment outcome, the TB treatment start date and the date the outcome was assigned were not missing. We used the Mann–Whitney/Wilcoxon rank-sum test to compare independent samples and non-parametric continuous data.

We present the results in four sections: 1) Accuracy of demographic information across sources; 2) Completeness of key TB-related variables across sources; 3) Accuracy of key TB-related variables and outcomes using measures of agreement, and 4) Accuracy of laboratory results, TB treatment outcomes and time to TB treatment outcome. The STROBE checklist for cross-sectional studies guided our documentation [23, 24].

## Ethical considerations

Ethical approval for the study was granted from the University of the Witwatersrand Human Research Ethics (Medical) Committee (protocol number M200702). This was a retrospective review of routine programmatic data and was exempt from collecting informed consent from patients.

## Results

### Sample description

Shown in Table 2, in total we reviewed 277 facility-based records (222 TIER.Net records/TB Treatment Records and 55 records from the TB Identification Register); in our first approach we paired 149/150 (99%) TIER.Net records to a paper TB Treatment Record. In our second approach we were also able to find a corresponding TIER.Net record from a starting point of a paper-based, TB Treatment Record for 73/75 records (97%). We found fewer records 55/75 (73%) in TIER.Net from the starting point of records from the TB Identification Register with similar findings across all sampled facilities. For the Register Review, over one quarter of records, 20/75 (27%) could not be paired to TIER.Net, in part due to a register not being available at four facilities. In the process of finding records, it appeared that patients who were HIV negative or had no laboratory confirmation of TB were not included in TIER.Net. For those that we could pair (n = 55), there were 20/55 (36%) TB-positive and 15/55 (27%) TB negative records. Of those 20 records without a TB-status on the register, 10/20 (50%) were HIV-positive.

**Table 2. Sample description as proportion of target.**

| Variable/Route of identification | All 15 facilities | Johannesburg Metro | Mopani District | Bojanala Platinum District |
|---|---|---|---|---|
| Gender* | | | | |
| Female | 108 (42%) | 36 (43%) | 39 (43%) | 33 (38%) |
| Male | 151 (58%) | 47 (57%) | 51 (57%) | 53 (62%) |
| Unknown | 18 | 6 | 5 | 7 |
| 1. TIER.Net to TB Treatment Record | 149/150 (99%) | 49/50 (98%) | 50/50 (100%) | 50/50 (100%) |
| 2. TB Treatment Record to TIER.Net | 73/75 (97%) | 25/25 (100%) | 25/25 (100%) | 23/25 (92%) |
| 3. Register Review | 55/75 (73%) | 15/25 (60%) | 20/25 (80%) | 20/25 (80%) |
| **Total** | **277/300 (92%)** | **89/100 (89%)** | **95/100 (95%)** | **93/100 (93%)** |

*unknown excluded from proportions.

## Accuracy of demographic information across sources

Of the 222 records we examined "1) TIER.Net to TB Treatment Record" and "2) TB Treatment Record to TIER.Net", there was 90% or more agreement for personal information like name, surname, and gender (Table 3). The agreement between records with date of birth and folder number were slightly lower at 91% (195/214) and 83% (177/214), respectively. For the TB Identification Register review (matching between the register and TIER.Net) there was 90% or more agreement for most of the demographic information compared, with surname and gender matching near perfect (98%).

## Completeness of key TB-related variables across sources

Table 4 compares the presence of key TB-related variables (e.g., registration type, patient category, treatment regimen and outcome) obtained from either the TB Treatment Record or from TIER.Net. For the sample of 222 records from TIER.Net, 96% (214/222) of TB Treatment Records could be located and analysed. The reporting of key TB-related variables was similar (p>0.05) across both sources for most variables except for TB treatment outcomes, site of disease, and smear microscopy or X-ray as a diagnostic test (p<0.05). An example of similarity: 76% (137/181) of TB cases from the TB Treatment Record versus 77% (158/205) from TIER. Net were newly registered. Conversely, a striking difference was found in TB treatment outcomes, 31% (51/165) were classified as cured in the TB Treatment Record while this was much lower in TIER.Net (11%; 24/214) (p<0.001).

**Table 3. Accuracy of demographic information, by source of information (n = 277).**

| Variable | Across 15 facilities |
|---|---|
| **TIER.Net to TB Treatment Record and TB Treatment Record to TIER.Net (n = 222)** | |
| Records Identified | 222 |
| Corresponding record not found[#] | 8/222 (3%) |
| Name-matched[*] | 197/214 (92%) |
| Surname-matched | 203/214 (95%) |
| Date of birth-matched | 195/214 (91%) |
| Gender-match | 207/214 (97%) |
| Folder number-match | 177/214 (83%) |
| Treatment start date—match | |
| **TB Identification Register to TIER.Net (excludes agreement to TB Treatment Record) (n = 55)** | |
| Identified | 55 |
| Matched with a TIER.Net Record | 51/55 (93%) |
| Name-matched | 47/51 (92%) |
| Surname-matched | 50/51 (98%) |
| Date of birth-matched | 45/51 (88%) |
| Gender-match | 49/51 (98%) |
| Folder number-match | 45/51 (82%) |
| TB Treatment Record not found | 23/55 (42%) |

[*]A match was identified if both sources of information reported identical information.

[#]We excluded records if the corresponding record could not be located (i.e., TIER.Net to TB Treatment Record; TB Treatment Record.

to TIER.Net or TB Identification Register to TIER.Net) and if the variable of interest was not present in both sources and could not be compared.

**Table 4. Completeness of data between clinic records and TIER.Net for key TB-related variables (n = 222).**

| Variable | TB Treatment Record: n (%) | TIER.Net: n (%) | P value* |
|---|---|---|---|
| **N** | 214 | 222 | |
| Referral forms | 78/214 (36%) | unavailable | - |
| **Registration type** | | | |
| Newly registered | 137/181 (76%) | 158/205 (77%) | 0.749 |
| Transferred/Moved-in (TMI) | 44/181 (24%) | 47/205 (23%) | |
| Missing | 33/214 (15%) | 17/222 (8%) | - |
| **Patient category** | | | |
| New | 187/201 (93%) | 202/213 (95%) | 0.295 |
| Relapse | 8/201 (4%) | 5/213 (2%) | |
| Retreatment after 1st line failure | 4/201 (2%) | 4/213 (2%) | |
| Retreatment after default | - | 2/213 (1%) | |
| Other (or multiple categories) | 2/201 (1%) | - | |
| Missing | 13/214 (6%) | 9/222 (4%) | - |
| **TB treatment regimen** | | | |
| Regimen 1 | 197/198 (99%) | 213/213 (100%) | 0.299 |
| Other | 1/198 (1%) | - | |
| Missing | 16/214 (7%) | 9/222 (4%) | - |
| **TB treatment start dates by quarter** | | | |
| Jan-Mar 2018 | 2/201 (1%) | 1/209 (1%) | 0.973¥ |
| Apr-May 2018 | 1/201 (1%) | 3/209 (1%) | |
| Jul-Sep 2018 | 1/201 (1%) | - | |
| Oct-Dec 2018 | 26/201 (13%) | 25/209 (12%) | |
| Jan-Mar 2019 | 36/201 (18%) | 39/209 (19%) | |
| Apr-Jun 2019 | 42/201 (21%) | 46/209 (22%) | |
| Jul-Sep 2019 | 70/201 (35%) | 71/209 (34% | |
| Oct-Dec 2019 | 23/201 (11%) | 24/209 (12%) | |
| Missing | 13/214 (6%) | 13/222 (6%) | |
| **Disease class** | | | |
| Extra pulmonary TB | 44/188 (23%) | 45/211 (21%) | 0.619 |
| Pulmonary TB | 144/188 (77%) | 166/211 (79%) | |
| Missing | 34/214 (16%) | 11/222 (5%) | |
| **Site of disease** | (available in S3 Table) | | - |
| Missing | 68/214 (32%) | 11/222 (5%) | - |
| **Labs/Tests (based on a checklist)** | | | |
| None | 15/214 (7%) | 13/222 (6%) | 0.696 |
| Smear Microscopy (TB sputum) | 144/214 (67%) | 164/222 (74%) | **0.039** |
| GeneXpert | 139/214 (65%) | 134/222 (60%) | 0.626 |
| X-ray | 10/214 (5%) | 34/222 (15%) | <**0.001** |
| Culture | 16/214 (7%) | 9/222 (4%) | 0.150 |
| LPA | 5/214 (2%) | 6/222 (3%) | 0.760 |
| Other | 16/214 (7%) | 7/222 (3%) | 0.054 |
| **TB treatment outcome** | | | |

(*Continued*)

**Table 4.** (Continued)

| Variable | TB Treatment Record: n (%) | TIER.Net: n (%) | P value* |
|---|---|---|---|
| Lost to follow-up (LTFU) | 4/165 (2%) | 19/214 (9%) | <0.001¥ |
| Died | 6/165 (4%) | 13/214 (6%) | |
| Transferred/Moved out | 9/165 (5%) | 17/214 (8%) | |
| Cured | 51/165 (31%) | 24/214 (11%) | |
| Treatment completed | 94/165 (57%) | 141/214 (66%) | |
| Rifampicin resistant | 1/165 (1%) | - | |
| Missing | 49 (23%) | 8 (4%) | - |
| **Presence of a treatment outcome date** | **157/214 (73%)** | **208/222 (94%)** | **<0.001** |
| Missing | 57/214 (27%) | 14/222 (6%) | - |
| **HIV status** | | | |
| Positive | 132/192 (69%) | 137/197 (70%) | 0.866 |
| Negative | 60/192 (31%) | 60/197 (30%) | |
| Unknown/Missing | 22/214(10%) | 25/222 (11%) | - |

*Proportions were compared across different sources using the chi-square test for proportions (or Fisher Exact test for sparse data)

¥ Fisher's exact.

**Regimen 1 is a 2-month intensive phase of Isoniazid (H), Rifampicin (R), Pyrazinamide (Z) and Ethambutol (E) and a 4-month continuation phase of Isoniazid (H), Rifampicin (R or Rif). Regimen 3 is for children and Other may include multi-drug resistant TB treatment–both outside of this paper's scope.

Abbreviations: ART–antiretroviral therapy, GeneXpert Mycobacterium tuberculosis (MTB)/rifampin (RIF), LPA—Line probe assay for detection of Mycobacterium tuberculosis, X-Ray—Energetic High-Frequency Electromagnetic Radiation.

Of note, for each of the key TB-related variables assessed, the missingness was less in TIER. Net compared to what was observed in the TB Treatment Record. Of particular significance is that almost one-third of TB treatment outcomes were not assigned in the TB Treatment record (27%; 57/214). Presence of a treatment outcome date was also significantly lower in the TB Treatment Record 157/214 (73%) as compared to what was in TIER.Net (208/222 94%; p<0.001). HIV status recording was among the most similar (p = 0.866). Missingness varied from: a 1% difference between the TB Treatment Record and TIER.Net for diagnostic lab test to a much larger difference of 27% for site of disease (32% missing in the TB Treatment Record compared to 5% in TIER.Net).

## Accuracy of key TB-related variables and outcomes using measures of agreement

We used the weighted Kappa statistic to calculate the agreement between key TB-related variables and outcomes (Table 5). The strength of agreement for all variables was more than moderate (Kappa >0.41), with four of the eight variables having good/substantial agreement (Kappa 0.61–0.80), but none of the variables indicated perfect agreement (Kappa >0.81). Of note, TB treatment outcomes had the poorest strength of agreement between sources (Kappa <0.5).

## Accuracy of laboratory results, treatment outcome and time to treatment outcome

We further interrogated the first GeneXpert result and the first TB smear result by creating a dichotomous variable (positive or negative) for these results obtained from the TB Treatment Record and from TIER.Net so we could visualize those results in a 2x2 table (S3 Table). We found that among patients that had GeneXpert or TB smear result, the agreement between

**Table 5. Measures of agreement between categorical variables captured in the clinic record and TIER.Net (n = 222).**

| Variable | Agreement | Expected agreement | Kappa statistic | Strength of agreement* |
|---|---|---|---|---|
| Registration type | 90.70% | 63.68% | 0.7439 | Good/substantial |
| Patient category | 94.95% | 88.38% | 0.5654 | Moderate |
| Disease class | 91.85% | 64.46% | 0.7706 | Good/substantial |
| Site of disease* | 90.04% | 75.83% | 0.5878 | Moderate |
| First GeneXpert test result | 92.00% | 65.74% | 0.7665 | Good/substantial |
| First smear result | 83.05% | 57.64% | 0.5999 | Moderate |
| HIV status | 84.06% | 47.17% | 0.6983 | Good/substantial |
| TB treatment outcome | 70.39% | 41.62% | 0.4929 | Moderate |

*We utilized linear weightings per STATA Kappa guidance and the multiple options of Site of disease.

**The value of Kappa was used to determine the strength of agreement; <0.20 poor, 0.21–0.40 fair, 0.41–0.60 moderate, 0.61–0.80 good/substantial and 0.81–1.00 very good/almost perfect.

sources was high. For the TB Treatment Record the first GeneXpert result, 75 positive results in TIER.Net matched the 76 positive results in the TB Treatment Record (99%). While 17 negative results in TIER.Net matched the 22 results in the TB Treatment Record (77% of cases). For first smear results: 24 positive results in TIER.Net matched the 31 positive results in the TB Treatment Record (77%). Additionally, 74 negative results in TIER.Net matched the 81 in the clinic records (91% of cases). In summation, when the result was "positive" in the clinical records it was more than likely "positive" in TIER.Net as well. Despite the differences we saw in Table 4, we saw agreement in terms of TB Outcomes, when we re-classified outcomes into successful (sum of treatment completed or cured) and unsuccessful (LTFU, died, treatment failure, transferred/moved out, rifampicin resistant, missing or unassigned outcome). TIER. Net was correlated with the TB Treatment Record when there was both a successful outcome 140/144 (98%) as well as an unsuccessful outcome 16/19 (84%) (S3 Table).

We compared time to TB treatment outcome by each source. Comparing the time to outcome, the median days (IQR) from start of TB treatment until outcome obtained from TIER. Net was 185 (168–216) while the median days (IQR) obtained from the TB Treatment Record records was higher 188 (180–231) (p = 0.252). While this result shows no difference in the time to TB treatment outcome between the TB Treatment Record and TIER.Net, the result should be interpreted with caution as it is limited to records that had both a TB start date and TB outcome date. From Table 4, recall 21/214 (10%) did not have a start date from the TB Treatment Record while 13/222 (6%) missing from TIER.Net; TB outcome dates were missing more often in the TB Treatment Record than in TIER.Net: 65/165 (40%) and 14/222 (6%) respectively; and TIER.Net had TB Treatment Outcome significantly more, about 20 percentage points higher.

## Discussion

This study provides insight into the data quality of the TB module in TIER.Net as it compares to the corresponding paper clinical file/records of TB treatment in public sector facilities in three South African districts. Our key finding was that TIER.Net was more complete in some key variables including TB treatment outcome, TB treatment outcome date and site of disease than the paper-based clinical record, the TB Treatment Record. In comparison, missing TB treatment start dates as well as HIV status recording had higher concordance. We also note some difficulty in finding TB records. In some cases, this was because of missing registers and also varied, not exclusively, on the HIV-status and TB-status of the records. Importantly,

TIER.Net, a part of the TB/HIV Information System (THIS) strategy (8,18), was designed as a patient management system to capture the cascade of HIV and TB care, including negative test results.

Our second main finding is that the agreement between TB Treatment Records and TIER. Net for demographic information (name, surname, date of birth, and gender) was high (>90%) and did not vary significantly across districts. The importance of this finding is the potential utility of linking this information to other data sources, where demographic information will be critical [25].

Our third main message is that the reporting (completeness) of key TB-related variables across the TB Treatment Record and the TB module in TIER.Net was similar for most variables (p>0.05), except for TB treatment outcomes, site of disease, and smear microscopy or X-ray as diagnostic tests (p<0.05). Where data was available to compare, "good/substantial" agreement was observed for registration type, disease class, first GeneXpert test result and HIV status. Of note, substandard agreement (Kappa <0.5) was observed for TB treatment outcomes between sources.

We attribute observing TIER.Net as more complete in terms of TB treatment outcomes, disease class and HIV status for the following reasons: 1) TIER.Net has built-in reports that can assist health care workers in the accuracy and completeness of records, an example of this is the "Outstanding TB Outcome List" in TIER.Net; 2) TIER.Net also has built-in internal logic controls such as only females can have a pregnancy status that are not possible on a paper record 3) outcomes may be recorded in clinical notes and not in designated fields in the standard TB stationery and are possibly misclassified during data capturing; and 4) there may be a time lag between when outcomes are assigned and when the file is captured in TIER.Net. This may provide data capturers with an opportunity to update TB treatment outcomes in TIER. Net from formal, such as a printed lab records, or undocumented sources, such as reading from a digital lab record, and not the paper TB Treatment Record.

HIV status was also more complete in TIER.Net and there was "good/substantial" agreement with the TB Treatment Record. However, our forth main finding is that HIV-status was not of optimum completeness as HIV status was missing in 30/214 (14%) of TB Treatment Records and 25/222 (11%) of TIER.Net records. This suggests further attention is needed on both good clinical practice and efforts to ensure HIV testing is well implemented and recorded.

Our fifth finding is that TIER.Net appears to be more complete for some key TB-related lab results. Of concern are some missing GeneXpert, culture and other tests (e.g. smear microscopy and LPA) in TIER.Net which requires further investigation. On further inspection, there was almost a perfect match (99%) in both sources if the GeneXpert was positive, while this was lower, 77%, when comparing positive smear results.

## Context from corresponding literature

Our work is aligned to some of the latest TB data in South Africa. Specifically, we report HIV-positive status of 132/197 (67%) in TIER.Net compared to 137/192 (69%) from the clinical records both of which is within 10% points to recently released national TB Prevalence Study, citing coinfection of HIV/TB at 59% [26]. The missingness we report from the TB Treatment Record is similar to what was reported by Jamieson et al. (2019) 11% of Disease Class (extra-pulmonary or pulmonary TB), but the missingness we report from TIER.Net was lower (5%) [14].

Historical assessments of ETR.Net data quality provide some helpful context: Podewils et al. (2015) evaluated accuracy, completeness and reliability of routine TB surveillance data in 2009

across two paper and four electronic sources (including the electronic TB register) [11]. They found that completeness varied across sources for some of the program indicators (e.g., HIV status) [27]. And that agreement also varied from: a high of kappa 0.94 for sex across sources; moderate for patient type (0.78), treatment regimen (0.79), treatment outcome (0.71); and poor for HIV status (0.33) [27]. Also well prior to the TB Module in TIER.Net, a study conducted in the Stellenbosch District of Western Cape Province indicated that 17% of TB cases with two smear-positive results were not recorded in the TB treatment registers [28]. Another study conducted at the Chris Hani Baragwanath Hospital in Johannesburg indicated that 21% of the patients referred to a clinic from the hospital did not attend any clinic in Soweto and were also not recorded in the TB treatment register [29].

As opposed to Podewils (2015) who reported the South African data as "often incomplete and inconsistent [27]," we are able to report two main positives: TB treatment data are available at the facility level and there is evidence of areas of improvement as well as areas of "good/substantial" data quality. Lastly, our data quality assessment suggests TIER.Net has adequately replaced ETR.Net. However, this is not without benefits and losses. Our findings do not show the presence of a previously strong and engaged sub-district management in the quality of TB data. The evidence of "good/substantial" data quality that we present here could suggest some of the fears of moving from away from ETR.Net to TIER.Net that Myburgh et al. explored have diminished [10].

This study suggests there is accurate and reliable data from the TB module in TIER.Net to update/replicate the estimates in Naidoo et al.'s work [20]. With the caveat that the current study has limited generalizability and that estimates of completeness and accuracy may vary across high and low volume TB facilities and other provinces across South Africa.

Another important consideration is the challenge of tracing TB patients in South Africa's very mobile population. We can report that about a quarter of the records we reviewed were transfer-in patients, and generally, we encountered complex referral pathways between hospitals and the primary healthcare settings in which we conducted our data collection. The challenge of the unique patient ID is worth mentioning as it is not being used throughout all the facilities we visited in Gauteng, Limpopo, and the North West despite assurances [30].

## Strengths and limitations

This work provides a valuable addition to the scanty literature of TB quality data; of four of the most recent publications on TB data quality in South Africa the most recent was covered data only from 2015–2016 [11, 13, 14, 31]. We conducted a robust and rapid data quality assessment across three different settings with Gauteng being a high-burden TB province, all during the first year of the COVID-19 pandemic. Our first limitation was our limited scope as we only reviewed TB data at the facility level and not across the entire health information system, specifically WebDHIS which is where national data are collated. Data access was also a limitation especially around missing TB Identification Registers. We used single data entry, while double data entry would have been a stronger data collection approach. We were unfortunately unable to link to patient-level data from National Health Laboratory Services patient-level data to TIER.Net records which would have enhanced the reliability of lab results. The study team was trained to collect data from designated fields in the standard TB stationary. Information recorded in the clinical notes (e.g., TB treatment outcomes or disease class) may have been overlooked by the study team, resulting in an underestimate of data completeness for some key TB-related variables. The retrospective nature of this study could have contributed to the low count of X-rays observed in the clinic record (5% versus 15% in TIER.Net). It is possible that laboratory results or ancillary reports placed in the TB Treatment Record could become

lost or misplaced over time. The relatively high missing records for HIV status in both the TB Treatment Record (14%; 30/214) and TIER.Net (11%; 25/222) is another area of concern. This gap requires further attention since an integrated patient management system may allow for easy identification of those with an unknown or missing HIV status. Lastly, our work did not assess data use to improve TB performance management which is a key driver to data quality and part of the purpose of TIER.Net through its automated reports and patient list [32].

## Conclusions

Improvements in data completeness of TIER.Net compared to the TB Treatment Record seems the most urgent area for improvement, especially around recording of TB treatment outcomes. Further operational research is needed to understand how data quality, data use and accurate reporting can be optimized. Lastly, we provide evidence that the TB module in TIER.net could be used to update the TB treatment cascade with seemingly improved data quality compared to the past.

## Supporting information

**S1 File. GW20-13 TB case identification register 1020 V9.**
(PDF)

**S2 File. GW-2012 2018 NDoH treatment record.**
(PDF)

**S1 Table. How we captured from TIER.Net.**
(DOCX)

**S2 Table. Site of disease details.**
(DOCX)

**S3 Table. 2x2 Tables comparing lab results.**
(DOCX)

## Acknowledgments

This manuscript is dedicated to Leila Reyes Hoffmann who passed away aged 11 years on 26 November 2021 from Ewing's sarcoma. As well as those we have lost from TB, COVID19, and other childhood cancers this year.

We extend our gratitude to Patricia Leshabana, Lerato Molapo, Kelebogile Kono, Lebogang Ngolele, Nkamoheleng Mokhesi and Jan Maboela for their diligent support and data caretaking during the study implementation. Additionally, our sincere thanks go to the Department of Health staff of participating clinics for accommodating the study, as well as patients attending these clinics as their records form the foundation of our findings.

## Author Contributions

**Conceptualization:** Joshua P. Murphy, Salome Charalambous, Lindiwe Mvusi, Sicelo Dlamini, Neil Martinson, Aneesa Moolla, Denise Evans.

**Data curation:** Joshua P. Murphy, Sharon Kgowedi.

**Formal analysis:** Joshua P. Murphy, Sharon Kgowedi, Constance Mongwenyana, Denise Evans.

**Funding acquisition:** Joshua P. Murphy, Salome Charalambous, Aneesa Moolla, Jacqui Miot.

**Investigation:** Joshua P. Murphy, Sharon Kgowedi, Vongani Maluleke, Daniel Letswalo, Constance Mongwenyana.

**Methodology:** Joshua P. Murphy, Jacqui Miot, Denise Evans.

**Project administration:** Joshua P. Murphy, Aneesa Moolla.

**Resources:** Joshua P. Murphy.

**Software:** Joshua P. Murphy.

**Supervision:** Joshua P. Murphy, Lezanie Coetzee, Salome Charalambous, Lindiwe Mvusi, Sicelo Dlamini, Neil Martinson, Aneesa Moolla, Jacqui Miot, Denise Evans.

**Validation:** Joshua P. Murphy, Denise Evans.

**Visualization:** Joshua P. Murphy, Lezanie Coetzee, Denise Evans.

**Writing – original draft:** Joshua P. Murphy.

**Writing – review & editing:** Joshua P. Murphy, Sharon Kgowedi, Lezanie Coetzee, Vongani Maluleke, Daniel Letswalo, Constance Mongwenyana, Pria Subrayen, Salome Charalambous, Lindiwe Mvusi, Sicelo Dlamini, Neil Martinson, Aneesa Moolla, Jacqui Miot, Denise Evans.

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
