## [Decision Letter · Decision Letter 0]

13 Jul 2022

PGPH-D-22-00374

Assessment of facility-based tuberculosis data quality in an integrated HIV/TB database in three South African districts

Dear Dr. Murphy,

Thank you for submitting your manuscript to PLOS Global Public Health. After careful consideration, we feel that it has merit but does not fully meet PLOS Global Public Health’s publication criteria as it currently stands. Therefore, we invite you to submit a revised version of the manuscript that addresses the points raised during the review process.

The manuscript has been evaluated by three reviewers, and their comments are available below.

The reviewers have raised a number of concerns that need attention. They request additional information on methodological aspects of the study and a few other details.

Could you please revise the manuscript to carefully address the concerns raised?

We look forward to receiving your revised manuscript.

Kind regards,

Julia Robinson

Executive Editor

Journal Requirements:

1. Please provide separate figure files in .tif or .eps format and remove the embedded figures within the manuscript file.

2. Since your data is not available for proprietary reasons, please explain via email why the data is not available. Please also include the contact information for the third party organization that should be contacted should other researchers want to request access to this data and please include the full citation of where the data can be found. We also request that you verify with us via email that any researcher will be able to obtain the data set in the same manner that the you have obtained it. If you feel you are unwilling or unable to adhere to this policy, please explain your reasons by return email and your exemption request will be escalated to the editor for approval. Your exemption request will be handled independently and will not hold up the peer review process, but will need to be resolved should your manuscript be accepted for publication. One of the Editorial team will be in touch if they require more information.

Additional Editor Comments (if provided):

Reviewers' comments:

Reviewer's Responses to Questions

**Comments to the Author**

1. Does this manuscript meet PLOS Global Public Health’s publication criteria? Is the manuscript technically sound, and do the data support the conclusions? The manuscript must describe methodologically and ethically rigorous research with conclusions that are appropriately drawn based on the data presented.

Reviewer #1: Yes

Reviewer #2: Yes

2. Has the statistical analysis been performed appropriately and rigorously?

Reviewer #1: Yes

Reviewer #2: Yes

3. Have the authors made all data underlying the findings in their manuscript fully available (please refer to the Data Availability Statement at the start of the manuscript PDF file)?

Reviewer #1: No

Reviewer #2: No

4. Is the manuscript presented in an intelligible fashion and written in standard English?

Reviewer #1: Yes

Reviewer #2: Yes

5. Review Comments to the Author

Reviewer #1: This manuscript describes a retrospective evaluation of data quality of the TB module in TIER.Net in three South African districts. I have few comments for the authors to consider and/or clarify.

Lines 67-69: “We report the quality of TB data captured in an electronic HIV/TB database, the Three Interlinked Electronic Register (TIER.Net), to contribute to this understanding.” This read as study aim/objective. I suggest moving this sentence to end of the introduction or rephrasing it.

Lines 109 -111: The authors conducted data quality assessment of the TB module in TIER.Net alongside a set of informal qualitative interviews to understand the strengths and weaknesses of the TB module in three districts of South Africa. What is meant by “set of informal qualitative interviews”? How these “set of informal qualitative interviews” were analysed? Why are the results not presented in the results section and discussed in the discussion section?

Lines 112 -117: The authors selected three districts in consultation with the South African NDoH Drug-sensitive (DS)-TB Directorate and National TB-Think Tank, however it is unclear why these three districts were selected.

Line 157: The authors used Stata and MS Excel for analysis. Why was MS Excel used for analysis?

Line 463: References need to formatted to follow a specific referencing style, especially WHO and governmental documents

Reviewer #2: The study compared TB records in South Africa across 3 sources - the TB treatment record or patient medical file; the TB Identification Register; and the TB module in TIER.Net – the electronic source.

The Goal was to evaluate data quality/ concordance of Tier.net data with the facility-based patient records and the ETR.net

Sampling:- logistical feasibility was described as the major reason for selecting the sites.

Data collection between October 2021 and December 2021;

Study population /patient records from October 18 -December 2019.

Objectives: Describe the sample ; the accuracy of demographic information – agreement between sources; completeness of Tb-related information; accuracy of key TB-related vars and outcomes – agreement between measurements; lab results agreement

Analytical methods: Kappa statistics for agreement, Mann–Whitney/Wilcoxon rank-sum test to compare independent samples and non-parametric continuous data; Chi-squared tests for independence of categorical variable.

Results: The distribution of demographic characteristics of patients across districts was fairly similar. The authors found Tier.net to be more complete for some key variables and they attribute that to some of the built-in features (e.g. logic checks) within the electronic db as compared to pare-based records.

General comment: This is a valuable contribution to TB-related research. Data quality [completeness and accuracy] is crucial for guiding TB programs. The manuscript is well-written overall, with some areas that need to be improved.

Comments

1. The abstract mentions three approaches that were used to make comparisons but these are not explicitly described in the methods section and it leaves the reader wondering. Could the abstract simply say something like “the comparisons were done with each of the three sources as a starting point. Throughout I found myself wondering what these “approaches” are.

2. Sampling – more details have to be provided about the selection of the 15 facilities. How were they situated across the 3 districts? Was a random approach used to select them?

3. Table 2 needs some clarification – it is not clear what the “Sum of DS-TB cases...” is. Is this the number of patient files available for this time period from which the study sample was selected? What is the relationship between the first row and the rest of the rows?

4. For key TB-related variables in Table 4, the authors conducted global tests of independence which indicates whether the patterns for reporting match between sources in general. This however does not give an exact evaluation of whether the values of these variables do match between sources. This is not as helpful as the results in Table 5 which lists agreement (the one-to-one correspondence check) between sources.

5. Within facility – a form of systematic sampling was used to select 10 records with treatment date closest to study dates of interest. It is mentioned that files that met eligibility criteria were selected but it is not clear what the eligibility criteria were.

6. PLOS authors have the option to publish the peer review history of their article (what does this mean?). If published, this will include your full peer review and any attached files.

**Do you want your identity to be public for this peer review?** For information about this choice, including consent withdrawal, please see our Privacy Policy.

Reviewer #1: No

Reviewer #2: **Yes: **Bareng A.S.Nonyane

---

## [Editor Report · Decision Letter 1]

6 Sep 2022

Assessment of facility-based tuberculosis data quality in an integrated HIV/TB database in three South African districts

PGPH-D-22-00374R1

Dear Mr Murphy,

We are pleased to inform you that your manuscript 'Assessment of facility-based tuberculosis data quality in an integrated HIV/TB database in three South African districts' has been provisionally accepted for publication in PLOS Global Public Health.

Best regards,

Julia Robinson

Executive Editor